# Robust continuous in vitro culture of the *Plasmodium cynomolgi* erythrocytic stages

Adeline C.Y. Chua[1,2,3,18], Jessica Jie Ying Ong[2,3,18], Benoit Malleret [1,4], Rossarin Suwanarusk[1,2], Varakorn Kosaisavee[4,5], Anne-Marie Zeeman[6], Caitlin A. Cooper[7], Kevin S.W. Tan [4], Rou Zhang[4], Bee Huat Tan[3], Siti Nurdiana Abas[3], Andy Yip[3], Anne Elliot[7], Chester J. Joyner[8,9], Jee Sun Cho[4], Kate Breyer[10], Szczepan Baran [10], Amber Lange[10], Steven P. Maher [7], François Nosten [11,12], Christophe Bodenreider[3], Bryan K.S. Yeung[3], Dominique Mazier[13,14], Mary R. Galinski[9,15], Nathalie Dereuddre-Bosquet[16], Roger Le Grand[16], Clemens H.M. Kocken[6], Laurent Rénia [1,4], Dennis E. Kyle [7], Thierry T. Diagana[3], Georges Snounou[13,14,16], Bruce Russell [2] & Pablo Bifani [1,3,4,17]

The ability to culture pathogenic organisms substantially enhances the quest for fundamental knowledge and the development of vaccines and drugs. Thus, the elaboration of a protocol for the in vitro cultivation of the erythrocytic stages of *Plasmodium falciparum* revolutionized research on this important parasite. However, for *P. vivax*, the most widely distributed and difficult to treat malaria parasite, a strict preference for reticulocytes thwarts efforts to maintain it in vitro. Cultivation of *P. cynomolgi*, a macaque-infecting species phylogenetically close to *P. vivax*, was briefly reported in the early 1980s, but not pursued further. Here, we define the conditions under which *P. cynomolgi* can be adapted to long term in vitro culture to yield parasites that share many of the morphological and phenotypic features of *P. vivax*. We further validate the potential of this culture system for high-throughput screening to prime and accelerate anti-*P. vivax* drug discovery efforts.

[1] Singapore Immunology Network, A*STAR, Singapore 138648, Singapore. [2] Department of Microbiology and Immunology, University of Otago, Dunedin 9054, New Zealand. [3] Novartis Institute for Tropical Diseases, Singapore 138670, Singapore. [4] Department of Microbiology and Immunology, Yong Loo Lin School of Medicine, National University of Singapore, Singapore 119077, Singapore. [5] Department of Parasitology and Entomology, Faculty of Public Health, Mahidol University, Bangkok 10400, Thailand. [6] Department of Parasitology, Biomedical Primate Research Centre, Rijswijk 2288, The Netherlands. [7] Center for Tropical and Emerging Global Diseases, University of Georgia, Athens 30602, USA. [8] Division of Pulmonary, Allergy, Critical Care & Sleep Medicine, Emory University, Atlanta 30322, USA. [9] Emory Vaccine Center, Emory University, Atlanta 30317, USA. [10] Laboratory Animal Services, Scientific Operations, Novartis Institutes for Biomedical Research, East Hanover 07936-1080, USA. [11] Shoklo Malaria Research Unit, Mahidol-Oxford Tropical Medicine Research Unit, Faculty of Tropical Medicine, Mahidol University, Mae Sot 63110, Thailand. [12] Centre for Tropical Medicine and Global Health, Nuffield Department of Medicine Research Building, University of Oxford Old Road Campus, Oxford OX3 7FZ, UK. [13] Sorbonne Universités, UPMC Univ Paris 06, CR7, Centre d'Immunologie et des Maladies Infectieuses (CIMI-Paris), Paris F-75013, France. [14] CIMI-Paris, INSERM, U1135, CNRS, Paris F-75013, France. [15] Division of Infectious Diseases, Department of Medicine, Emory University, Atlanta 30322, USA. [16] CEA-Université Paris Sud 11-INSERM U1184, Immunology of Viral Infections and Autoimmune Diseases (IMVA), IDMIT Department, IBJF, DRF, Fontenay-aux-Roses 92265, France. [17] Faculty of Infectious and Tropical Diseases, London School of Hygiene & Tropical Medicine, London WC1E 7HT, UK. [18]These authors contributed equally: Adeline C.Y. Chua, Jessica Jie Ying Ong. Correspondence and requests for materials should be addressed to B.R. (email: b.russell@otago.ac.nz) or to P.B. (email: micpb@nus.edu.sg)

The development of a protocol for the routine continuous in vitro culture of *Plasmodium falciparum* in 1976[1] released malaria researcher from the reliance on in vivo observations. This led to the major fundamental and translational advances in all aspects of the life cycle of this parasite that is responsible for the highest mortality rates globally. Recent recognition that the widespread species *P. vivax* causes substantial morbidity[2] made it imperative to devise efficient means to control it. Furthermore, effective measures that could lead to its elimination would require the development of novel drugs to eliminate the hypnozoite, the dormant liver form responsible for relapses that also characterise *P. vivax* infections, as well as strategies to thwart transmission. However, research on *P. vivax* remains severely hampered because, to date, attempts to maintain this parasite in routine in vitro blood cultures have been hindered by the strict restriction to invasion of reticulocytes, a minor short-lived fraction of peripheral blood. Historically, a parasite of Southeast Asian macaques, *P. cynomolgi*, has been used as the favoured model for *P. vivax*. The remarkable morphological and biological similarities between these two parasite species are now known to extend to their genetic make-up[3,4]. The hepatic stages of malaria parasites were first discovered using *P. cynomolgi*[5] as was the hypnozoite[5–7]. Indeed, *P. cynomolgi* was central to the development of primaquine[8] and remains so in the search for novel anti-relapse compounds both in experimental infections[9,10] and as a basis for the recently developed in vitro-cultured hypnozoite model[11–13].

In the last decade, high-throughput screens based on cultured *P. falciparum* or on those of the hepatic stages of the rodent parasites *P. berghei* or *P. yoelii*, as reviewed in Hovlid and Winzeler[14], have enriched the drug discovery pipeline with a wealth of promising novel lead compounds. However, such screening strategies are precluded for *P. vivax* because of the limited availability of infected blood from patients, an obstacle that would be circumvented should in vitro-cultured *P. cynomolgi* be available. There were two reports of continuous in vitro culture of the blood stages of two strains of *P. cynomolgi* (Berok in one, and Vietnam in the other) in the early 1980's[15,16]. Given the regular and onerous use of *P. cynomolgi* in macaques for biological and drug discovery programmes, it was surprising that these early observations were not exploited further.

Following careful assessment of various culture conditions, we describe here the robust continuous cultivation of the blood stages of *P. cynomolgi* lines derived from the Berok strain. We show that the in vitro-cultured *P. cynomolgi* (from ex vivo or cryopreserved stocks) retain the key characteristics that these parasites share with *P. vivax*. We further demonstrate that these parasites are suitable for high-throughput screening for antimalarial compounds.

## Results

**Propagation of *P. cynomolgi* strains**. Three *P. cynomolgi* lines were available to us: the Berok strain cryopreserved in 2003 from *Aotus trivirgatus* (initially isolated in 1964), the B strain (*P. cynomolgi bastianelli*, initially isolated in 1959) and the M strain (*P. cynomolgi* Mulligan strain first isolated in the 1930s). Separate *Macaca fascicularis* monkeys were successfully infected with one or other of these lines (see the Methods section), and blood samples were collected to initiate cultures. We conducted preliminary experiments using the Berok strain, to test various culture conditions, media and materials in short-term cultures initiated from cryopreserved stocks prepared from infected monkey blood (Fig. 1a). This allowed us to define an optimal working protocol that was then tested on Berok strain parasites from different animals, and on B and M strain parasites

freshly collected from infected macaques. Blood was collected when most parasites were mature (late trophozoites and schizonts), and depleted of leucocytes before enrichment of the mature parasites on a Percoll gradient. In vitro cultures were initiated at parasitaemia > 1% (50,000 parasites/μL) and monitored daily by microscopic examination of Giemsa-stained smears. In repeated independent experiments, the parasitaemia of the B and M strain cultures declined to become undetectable within a few days (Fig. 1b). By contrast, erythrocyte invasion was observed in the cultures initiated with the Berok strain obtained from different monkeys (K2, K3 and K4). Nonetheless, the multiplication rates of the K2-initiated and K3-initiated cultures were low by comparison with that of the K4-derived parasites. Thus, we opted to continue subsequent work with the K4 culture. This initial Berok K4 line culture, had a multiplication rate ranging from twofold to fourfold over more than five cycles, such that regular dilution of the culture became necessary. The K4 line was used to constitute working cryopreserved stocks to conduct the studies described below.

**Biological characteristics of in vitro-cultured Berok K4**. Successful propagation of the Berok K4 line was maintained for up to 180 days in cultures initiated with thawed parasites. Culture conditions were further refined, with robust growth best observed under reduced oxygen environment (5% $CO_2$, 5% $O_2$ and 90% $N_2$) to reach parasitaemias beyond 10% (Fig. 1c). Growth rates were unaffected by flask configuration (24-well plate, six-well plate, T25 or T75 flasks). Successful in vitro maintenance of the Berok K4 stabilates was obtained independently in six additional laboratories (the National University of Singapore, the University of Otago in New Zealand, the University of Malaya in Malaysia, the Shoklo Malaria Research Unit in Thailand, the Novartis Institute for Tropical Diseases, Emeryville, California and at the Université Pierre et Marie Curie in France). Genotyping of the *reticulocyte binding protein* gene complex, whose types and copy number vary between strains[3,17–19], confirmed that the K4 line parasites were indeed derived from the original Berok stock (Supplementary Fig. 1). The morphology of the parasites sampled at 2 hourly intervals for 48 h (Fig. 1d) was indistinguishable from that observed or previously described from infected animals, with eight to sixteen merozoites observed in fully matured schizonts around 46–48 h. Although macrogametocytes and microgametocytes were intermittently observed from day 6 on, their levels remained low and did not exceed 0.01% gametocytaemia. Culture condition modifications that would lead to the higher sexual stage production in vitro are currently being tested.

**In vitro-cultured Berok K4 line retains transmission ability**. In vitro-cultured Berok K4 line parasites were used successfully to infect a naive intact *M. mulatta* monkey, where asexual and sexual stages were observed 6 days post infection (dpi) and thereafter. *Anopheles stephensi* mosquitoes fed on infected blood collected at 11 and 12 dpi (0.85% and 1.4% parasitaemia, respectively) showed oocysts a week later (feed 11 dpi: in 40% of mosquitoes, average of 3.5 oocysts/mosquito; feed 12 dpi: in 90% of mosquitoes, average of 37.6 oocysts/mosquito). Eighteen days after feeding, lots of 100,000 sporozoites isolated from salivary glands (feed 12 dpi: ca. 31,000 sporozoites per mosquito) were successfully used to infect two naive *M. mulatta* monkeys by intravenous inoculation (Fig. 2a), and the course of the infections was monitored by microscopy. Both monkeys became patent 11 dpi, and were later administered a 5-days chloroquine regimen to eliminate blood-stage parasites in order to monitor for relapses as previously described[20]. Monkey 1 relapsed 31 dpi and 52 dpi, after which radical cure was administered (chloroquine and primaquine), while the second remained negative for the whole

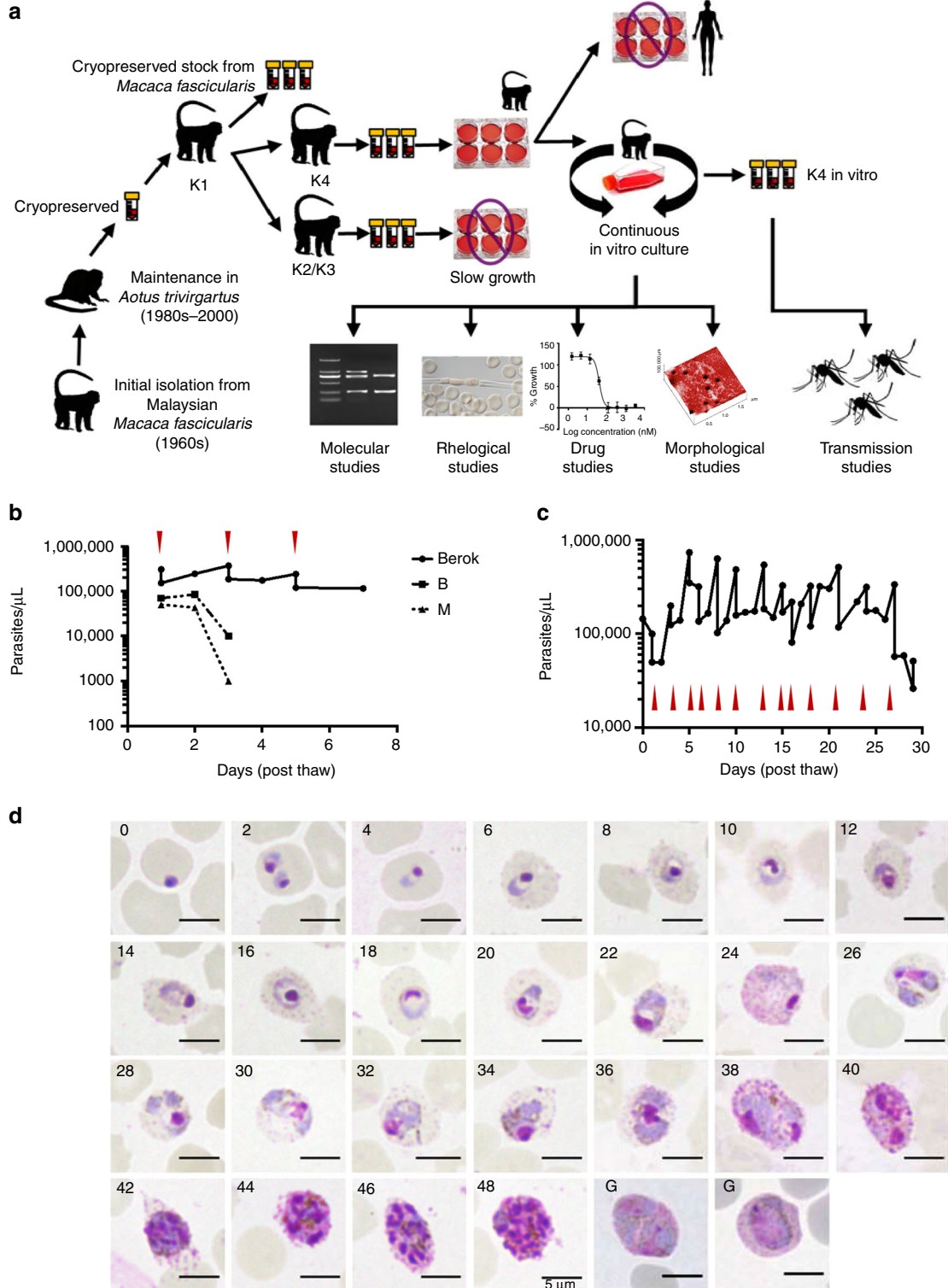

**Fig. 1** Adaptation of *P. cynomolgi* Berok K4 line to continuous culture. **a** Schematic diagram of the successful adaptation of *P. cynomolgi* (Berok) from in vivo to in vitro culture. **b** Pilot ex vivo culture of *P. cynomolgi* Berok, B and M strain. The in vitro-cultured Berok K4 had to be sub-cultured at days 1, 3 and 5 due to robust growth, in contrast to the M and B strains where parasitaemias decreased to undetectable levels after day 3. **c** *P. cynomolgi* (Berok) in vitro culturing was further optimised to enable substantial multiplication (up to tenfold increase) that necessitated frequent dilution of the cultures when high parasitaemias were reached (red arrowheads). **d** Mature schizonts of culture adapted *P. cynomolgi* Berok K4 were enriched and allowed to re-invade fresh red blood cells that were then monitored every 2 h to document the complete asexual erythrocytic cycle in vitro. Scale bar represents 5 μm

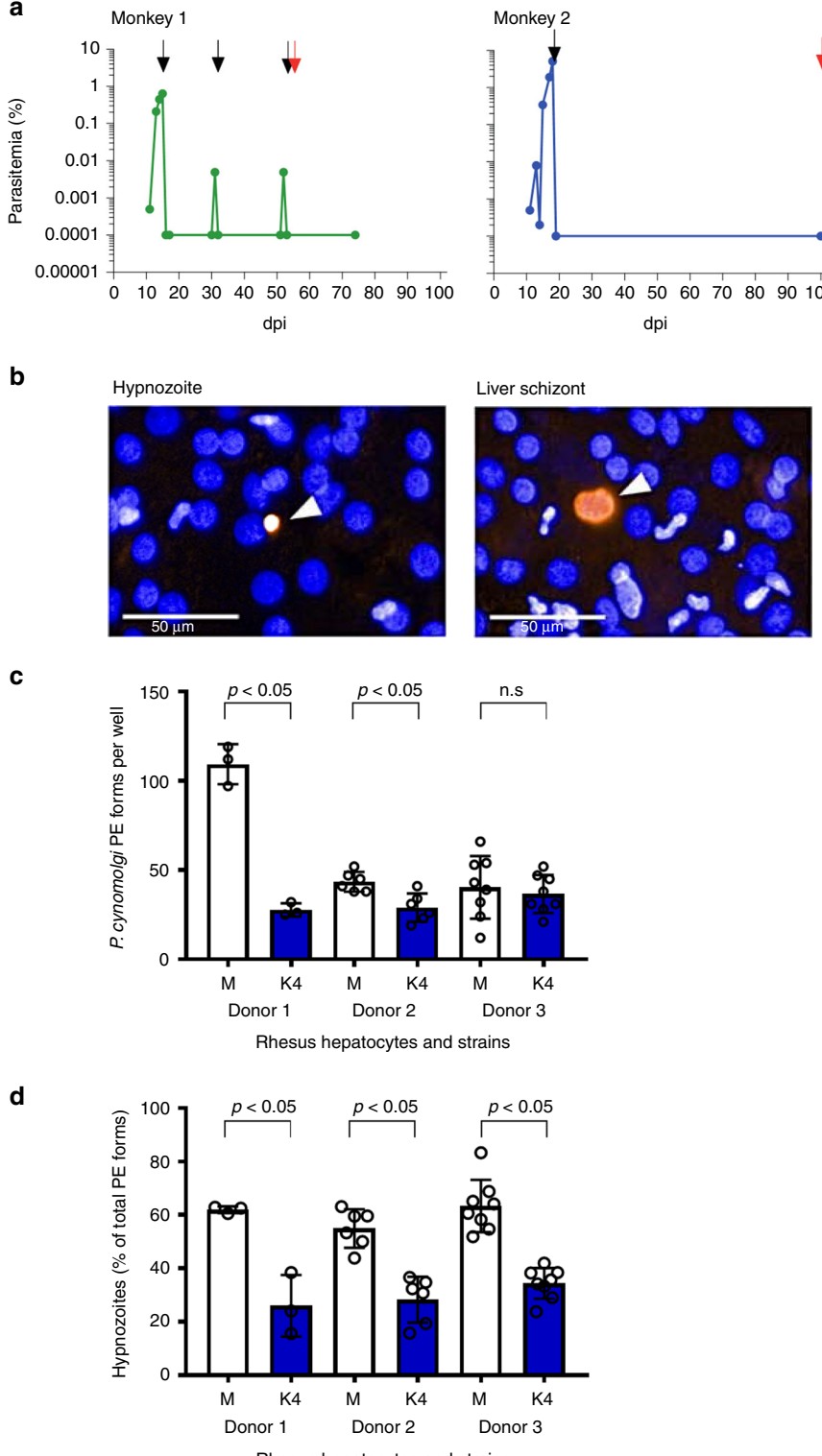

duration of follow-up (102 dpi). The time to patency was similar to that noted for *P. cynomolgi* M strain infections (using the same sporozoites inoculum and infection route), as was that for the first relapse (around day 27.5 ± 3.25 dpi in eight monkeys infected by the M strain[20]). Usually, monkeys infected with *P. cynomolgi* M strain relapse at least once, regardless of inoculation dose, which contrasted with the relapse episodes noted in only one of two rhesus monkeys inoculated with the *P. cynomolgi* Berok K4 sporozoites. Sporozoites were also used to infect in vitro-

cultured *M. mulatta* primary hepatocytes, as previously descri-bed[11], and 6 days later schizonts and uninucleate parasites (possibly hypnozoites) were detectable (Fig. 2b). The average infectivity of the sporozoites derived from the *P. cynomolgi* Berok K4 line was lower, around 16 pre-erythrocytic (PE) forms per 10,000 inoculated sporozoites, than that of those derived from the *P. cynomolgi* M strain, around 23 PE forms per 10,000 inoculated sporozoites (Fig. 2c). Enumeration of the two hepatic forms suggested that the *P. cynomolgi* Berok K4 line generated half the

**Fig. 2** Transmission study from *P. cynomolgi* Berok K4 continuous culture. **a** In vivo blood-stage parasitaemia in two rhesus monkeys infected with 100,000 *P. cynomolgi* Berok K4 sporozoites (because of the use of a log scale for parasitaemia, negative smears are shown as 0.0001% parasitaemia). Both monkeys became blood-stage patent on day 11 post infection (dpi). Arrows indicate drug treatment (black arrows: 5-day chloroquine treatment, red arrow: 7-day primaquine treatment). Monkey 1 was bled for stocks on day 15, and relapsed (measured as thin smear positivity) on days 31 and 52, after which it was treated with chloroquine and primaquine. Monkey 2 was bled on day 19 dpi, and did not relapse during the follow-up period of 102 dpi, after which it was treated with primaquine. **b** In vitro infection of primary rhesus hepatocytes with *P. cynomolgi* Berok K4 sporozoites produced both hypnozoites (left panel) and developing liver-stage schizonts (right panel). Cultures were fixed at day 6 dpi, and stained with anti-PcHsp70 and a secondary antibody labelled with Alexa 568 fluorescent dye. Nuclei were stained with DAPI. An average of 16 PE forms per 10,000 inoculated *P. cynomolgi* Berok K4 sporozoites were observed. Scale bar represents 50 μm. **c** The total PE forms of in vitro infection rate of various primary rhesus hepatocytes with *P. cynomolgi* M strain sporozoites and *P. cynomolgi* Berok K4 line sporozoites. **d** The percentage of hypnozoites observed in vitro using primary rhesus hepatocytes from different donors infected with *P. cynomolgi* M strain or *P. cynomolgi* Berok K4 line sporozoites. The data (**c** and **d**) were analysed using the Welch's *t* test with the significance level set at $P < 0.05$. The histograms represent means ($n = 3$), and the error bars the standard error of the mean (SEM) of replicates

number of hypnozoites (around 30%) compared with that produced by the *P. cynomolgi* M strain (around 60%) (Fig. 2d). Further studies are required to ascertain whether the failure of one of the monkeys infected with the Berok K4 line to relapse within the 102-day observation period reflects the lower numbers of uninucleate forms observed in vitro, or to a delayed relapse pattern where the first relapse occurs many months after sporozoite inoculation, or to residual chloroquine that might have suppressed a first relapse.

**Nanostructure and rheology of Berok K4 line-infected RBCs.**
*P. cynomolgi* Berok K4 dramatically alters the nanostructural and rheological properties of the infected red blood cell (iRBC) in a manner similar to that observed for *P. vivax*. Caveolae (~90 nm-diameter openings) appear on the surface of the infected RBC's (Supplementary Fig. 2). While these caveolae, which are generally associated with vesicle complexes, have the same dimensions as those noted in *P. vivax*, these are present in significantly lesser densities (Fig. 3a–d), our data on caveolae agree with previous in-depth analyses conducted on *P. cynomolgi* Berok[21]. The *P. cynomolgi* Berok K4-infected RBC increases in size with maturation (rings ~5 μm and schizonts ~8 μm) with rosettes forming ~20 h post invasion (Fig. 3e, f), a timeline similar to that previously noted for *P. vivax*[22]. It is interesting that rosettes had not been observed in earlier studies on the B and M strains[23]. The *P. cynomolgi* Berok K4 rosettes formed highly stable adhesive bonds requiring around 400 pN (piconewton) to affect a separation (Fig. 3g, h), an adhesive force similar to that recorded for *P. vivax*[24].

**High-throughput drug susceptibility assay using the Berok K4 line.** We opted to validate the SYBR green I proliferation assay[14,25] for use with the *P. cynomolgi* Berok K4 line, because it is used routinely for anti-*P. falciparum* drug screening, to establish dose response and single-point screens based on 96- and 384-well-plate assay formats (Fig. 4a–c). Using the BioTek Synergy™ 4 hybrid microplate reader, the range of the 72 h fluorescent read-out in the SYBR green I proliferation assay showed a linear correlation with parasitaemia in the range of 0.3–2%. Culture medium was supplemented with 20% macaque (Mf) serum for the Berok K4 line, whereas the routine 0.5% AlbuMAX was used for *P. falciparum*. Nonetheless, $Z'$ prime values from the average of 16 DMSO wells (negative control) and 16 mefloquine wells (positive control) ranged from 0.8 to 0.9 (Supplementary Fig. 3). In order to minimise the use of materials from naive macaques for culture, initial parasitaemia of 0.5 and 2% were chosen. A SYBR green I proliferation assay was performed in a 384-well-plate format using a set of reference compounds and selected compounds from the Malaria Box (Medicines for Malaria Venture, Switzerland) (Fig. 4a, b). The reference compounds included the licensed antimalarials

chloroquine, lumefantrine, pyrimethamine, artemisinin, atovaquone and artesunate, as well as three drug candidates: the phosphatidylinositol-4-OH kinase (PI(4)K) inhibitor KDU691, the imidazolopiperazines KAF179 and the spiroindolone KAF246 (KAF179 and KAF246 are analogues to KAF156 and KAE609 currently in phase 2b clinical trials[26]). The IC$_{50}$ values measured for both species were broadly concordant, though *P. cynomolgi* proved more sensitive to artesunate, artemisinin and atovaquone (Fig. 4a). The assay was also conducted for 38 in-house synthesised compounds from the Malaria Box (Fig. 4b), and identified compounds that had differential inhibitory activity against the two parasites species. For example, MMV000563, MMV007839 and MMV008294 were highly active against *P. falciparum*, but not *P. cynomolgi*.

In order to evaluate whether the type of culture supplement (serum or AlbuMAX) influences inhibitory activity, the Pathogen Box chemical library (Medicines for Malaria Venture, Switzerland) was screened in a 384-well format as a single-point assay at 10 μM (Supplementary Fig. 4) using *P. cynomolgi* (20% Mf serum) or *P. falciparum* (20% human serum or 0.5% AlbuMAX). The Pathogen Box comprises 400 compounds, of which 125 are antimalarial tool–compounds, 26 are reference compounds while the rest include compounds active against tuberculosis ($n = 116$), kinetoplastids ($n = 70$), helminths ($n = 32$), cryptosporidiosis ($n = 11$), toxoplasmosis ($n = 15$) and dengue ($n = 5$) (http://www.pathogenbox.org/). Assay data for *P. cynomolgi* in 20% Mf serum (Column A) proved to be highly comparable with that obtained for *P. falciparum* in 20% human serum (Column B), but both assays differed significantly from the *P. falciparum* assay performed in 0.5% AlbuMAX (Column C). This observation was not surprising given the high protein content of 20% serum-supplemented media with the likely consequent effect on protein binding. As above, the rate of inhibition of nine antimalarial compounds (MMV676380, MMV023388, MMV026550, MMV007625, MMV023949, MMV007638, MMV676442, MMV006833 and MMV020289) were significantly different for both parasite species in the presence of 20% serum (Fig. 4c). Overall, the SYBR green I proliferation assay demonstrated robust reproducibility, at single point and for dose response, for both *Plasmodium* species in serum-supplemented cultures.

**Schizont maturation assays in the Berok K4 line and *P. vivax*.**
Cultured *P. cynomolgi* is to be ultimately used as a surrogate for *P. vivax* in drug sensitivity assays. Therefore, we carried out the standard schizont maturation assay for *P. vivax*, using tightly synchronised Berok K4 ring-stage cultures as well as several clinical *P. vivax* isolates. The parasites were seeded on plates containing serially diluted chloroquine and parasite development was assessed about 44 h later by flow cytometry (Fig. 5a). In parallel, the SYBR green I proliferation assay was carried out for the same drug using

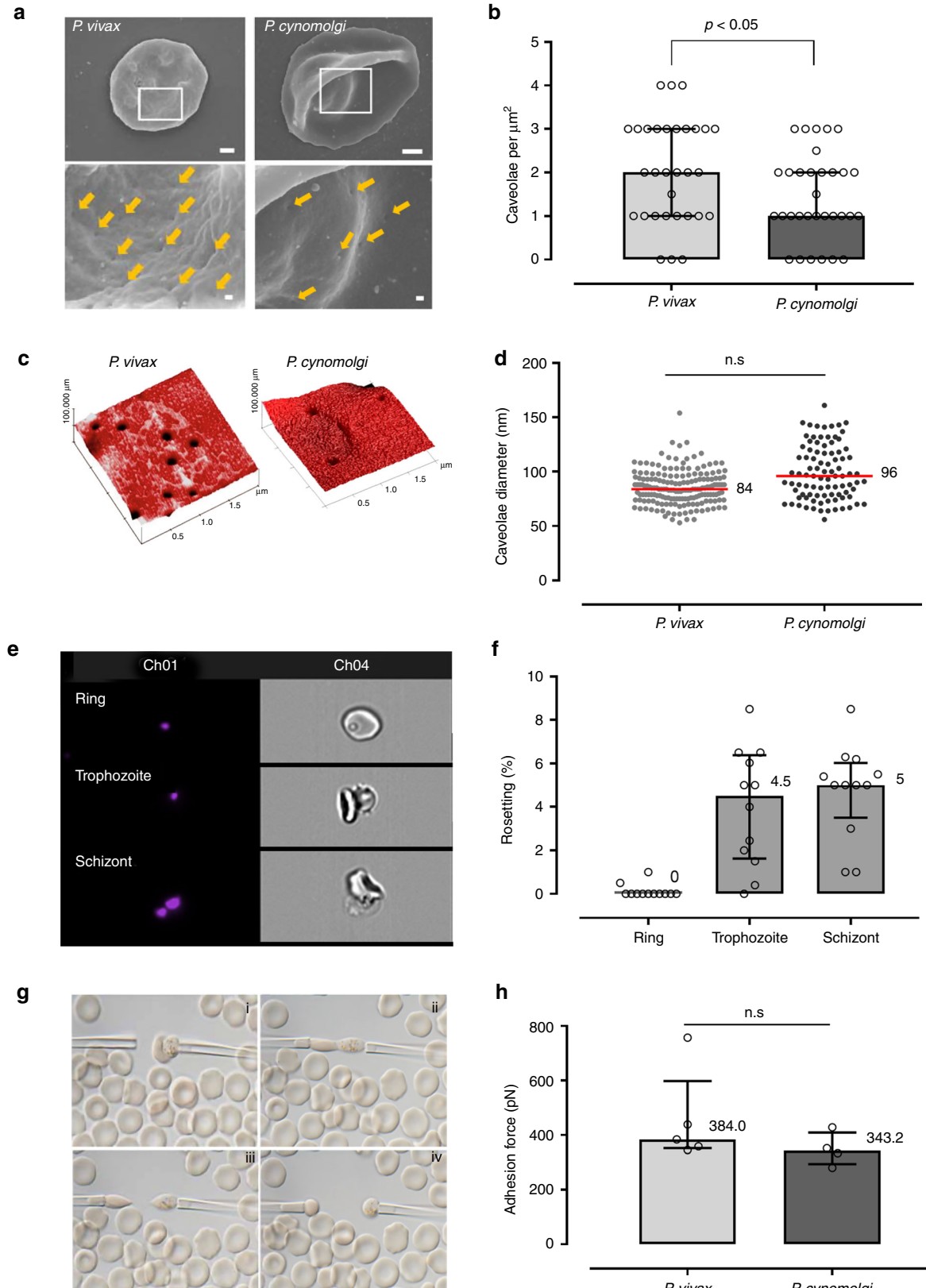

the Berok K4 line. Each dot represents individual *P. vivax* clinical isolates and replicates for the Berok K4 line, respectively. The IC$_{50}$ values for chloroquine were similar for *P. cynomolgi* in both assays (Fig. 5b, c), and equally comparable with the IC$_{50}$ values (~50 nM) obtained from the *P. vivax* clinical isolates (Fig. 5d).

## Discussion

The control of the globally distributed *Plasmodium vivax* became, over the last decade, a priority for the malaria community. Indeed, the biological characteristics that distinguish this species from *P. falciparum*, in particular the propensity to relapse and its

**Fig. 3** Phenotypic and rheological characterisation of the Berok K4 line from in vitro culture. **a** *P. cynomolgi* Berok K4-infected RBCs exhibit caveolae structures (yellow arrows) that are similar to those in *P. vivax*-infected RBCs (scanning electron microscopy, scale bars represent 1 μm and 100 nm for area shown at higher magnification in white box). **b** An atomic force microscope scan of trophozoite-infected human blood cells revealed caveolae occurred at lower frequency when compared with *P. vivax*. **c**, **d** The median (+/- IQR) dimensions of these caveolae were similar (*P. vivax* $n = 177$, *P. cynomolgi* $n = 91$). **e**, **f** Amnis flow imaging clearly shows that the mature erythrocytic stages *P. cynomolgi* Berok K4; readily formed rosettes with uninfected red blood cells, which are also a key feature *P. vivax* ($n = 5$). **g**, **h** A dual micropipette aspiration method was used to demonstrate the rheological stability of the *P. cynomolgi* Berok K4 rosettes ($n = 5$). As observed in *P. vivax*, *P. cynomolgi* rosettes are tightly attached and the cells require around 400 pN to disrupt the adhesion. The non-parametric data resented in **b**, **d**, **f** and **h** were analysed using the Mann–Whitney U test with the significance level set at $P < 0.05$. The histograms and lines on box plots and scatter plots represent medians, and the error bars the interquartile range (IQR)

consequences and the increased vectorial capacity, are a significant challenge to current elimination efforts[27]. Moreover, the increasing reports of resistance to chloroquine and the lack of an anti-hypnozoite drug that could be mass-deployed have made it a priority to seek novel compounds active against this parasite. Furthermore, efforts to develop a vaccine against *P. vivax* significantly lag behind those devoted against *P. falciparum*. The inability to culture *P. vivax* in the laboratory has in equal measures thwarted efforts to conduct functional studies and to screen for novel lead compounds targeting *P. vivax*. Thus, investigations of all aspects of *P. vivax* and the infection it causes are restricted to field and clinical observations, and to the few samples that can be obtained from patients prior to treatment. The high cost and ethical limitations inherent to the use of primate models, *P. vivax* lines adapted to New World non-human primates, or *P. cynomolgi* in macaques, further constrain the scope of any investigations. The availability of in vitro-cultured *P. cynomolgi* makes this suitable model for *P. vivax* available to a broad range of researchers and opens the way to apply genetic manipulation technologies that are as yet precluded for *P. vivax*.

The first protocol for in vitro cultivation of the erythrocytic stages of malaria parasite was made for *P. falciparum* and *P. vivax* in 1912[28], though growth was restricted to a few cycles. Sustained multiplication over extended periods (months), i.e., continuous cultures, proved elusive and was finally achieved only in 1976 for *P. falciparum* [1,29] after sustained efforts spanning three decades. Within a few years, the continuous cultivation of four macaque parasite species, *P. knowlesi*, *P. cynomolgi*, *P. inui* and *P. fragile* were reported[30]. Whereas cultures of *P. knowlesi* were later exploited to conduct genetic and biological studies[31,32], none appear to have been similarly employed for *P. cynomolgi*, despite the evident benefits that this could have provided over the years to researchers investigating *P. vivax*. The fastidious testing of culture conditions that we carried out and our subsequent observations on the inability of some strains to grow in culture might in part explain this gap. Human serum to supplement the culture medium (successfully used in the initial publication from 1981) proved unsuitable in our hands, despite testing of numerous batches. Moreover, though collected from naive animals, then prepared and stored in a standard manner, the serum and in some cases the red blood cells collected from only some but not all naive *M. fascicularis* proved suitable for sustained parasite multiplication.

It is not clear at present why establishment in vitro under the adopted standard conditions appears to be limited to the Berok line (the *P. cynomolgi* M or B strains failed to thrive, despite numerous attempts). This phenomenon is not novel or peculiar to *P. cynomolgi*, as similar variable success in adapting field-collected *P. falciparum* isolates has long been observed. The successful adaptation to in vitro culture remains elusive and multifaceted. It is well documented that the *rbp* genes are different in the various strains of *P. cynomolgi*[3,17]. It will be useful to explore whether the fact that only the Berok strain, but neither the B nor M strain, possesses the *rbp1b* gene played a role. This

gene is present in *P. vivax*[3] though this species remains uncultivatable in vitro. Clearly, the mechanisms of merozoite invasion are complex, and will require detailed investigations. Moreover, red blood cell tropism might have played a role. In a recent study by Kosaisavee et al.[33], the *P. cynomolgi* B strain equally invaded all types of monkey erythrocytes, while it was restricted to Duffy-positive human reticulocytes.

The Berok strain, initially derived from a wild *M. nemestrina* collected in peninsular Malaysia in the early 1960s, was maintained thereafter by blood and/or sporozoite inoculation primarily in *M. mulatta* and later also in *Aotus* monkeys[34]. This strain has not been cloned, and it is possible that it might harbour diverse parasite genotypes which could vary in their proportion over the course of infection and or in different hosts. It is at present unknown whether the derived Berok parasites collected from the various monkeys (K2–K4) differed in the proportion of parasites that can rapidly adapt to thrive in culture, possibly as genetically distinct lines from an otherwise heterogeneous *P. cynomolgi* population, or as phenotypic variants selected from an otherwise homogeneous population. Elucidation of the molecular basis for this differential growth could provide fundamental insights on the biology of blood-stage parasites. It is likely that lines from other *P. cynomolgi* isolates and strains, or indeed from those of other macaque parasite species, would also be amenable to blood-stage cultivation, provided that the initial failures are met with perseverance.

It is our intention to facilitate the dissemination of the Berok K4 line from *P. cynomolgi*, a species proven as an excellent surrogate for *P. vivax*, so as to broaden the scope of investigations that in vitro-cultured parasites would allow. Routine cultivation is easily initiated from cryopreserved stocks using the conditions and methodology that were optimised to enable robust growth and large-scale production, should this be needed. We have shown that in vitro-produced Berok K4 parasites are indistinguishable morphologically from in vivo and ex vivo parasites, and that they retained their infectivity to monkeys and subsequently to mosquitoes to generate infective sporozoites. The presence of caveolae, and the occurrence of rosettes, provides additional evidence that this parasite species also shares the characteristic morphological, phenotypical and rheological alterations observed for *P. vivax*-infected red blood cells.

The last 10 years have witnessed significant successes in the search for new antimalarial compounds, with 17 new drug candidates developed since 2010[35,36]. The key contribution was the development of a standard *P. falciparum* asexual blood-stage SYBR green I proliferation assay adapted to automated screening technologies[14]. Such strategies are not possible for *P. vivax*, and high-throughput screening of compounds would require a prohibitively large number of *P. cynomolgi*-infected macaques. We validated the potential of the *P. cynomolgi* Berok K4 line cultures to extend high-throughput screens to identify lead compounds active against *P. vivax*. It was interesting to note that some of the compounds from the Malaria Box and the Pathogen Box used for the validation were inhibitory to one, but not the other of the

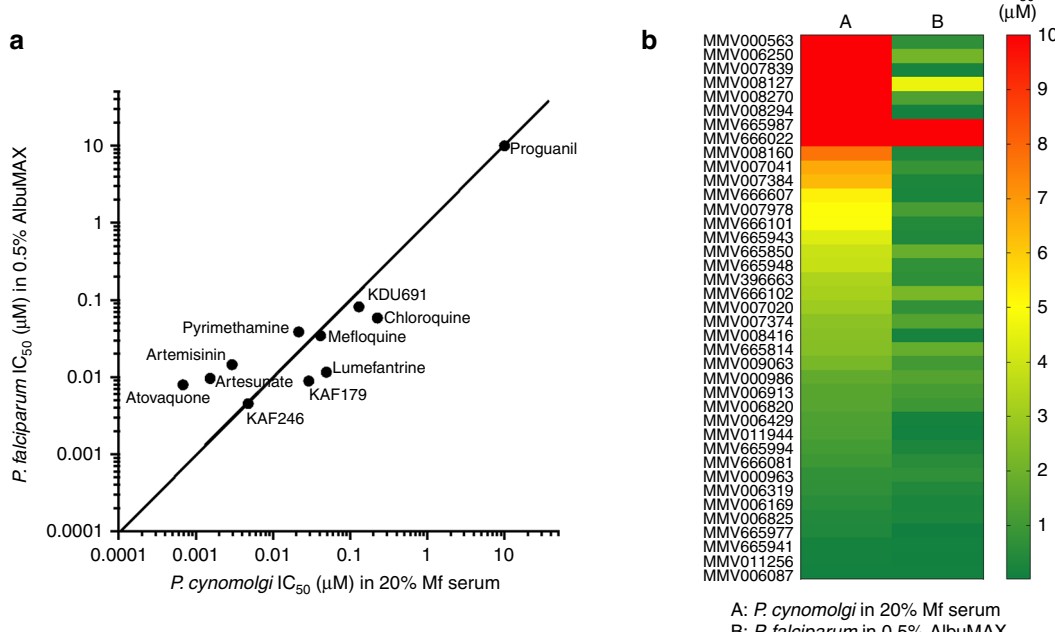

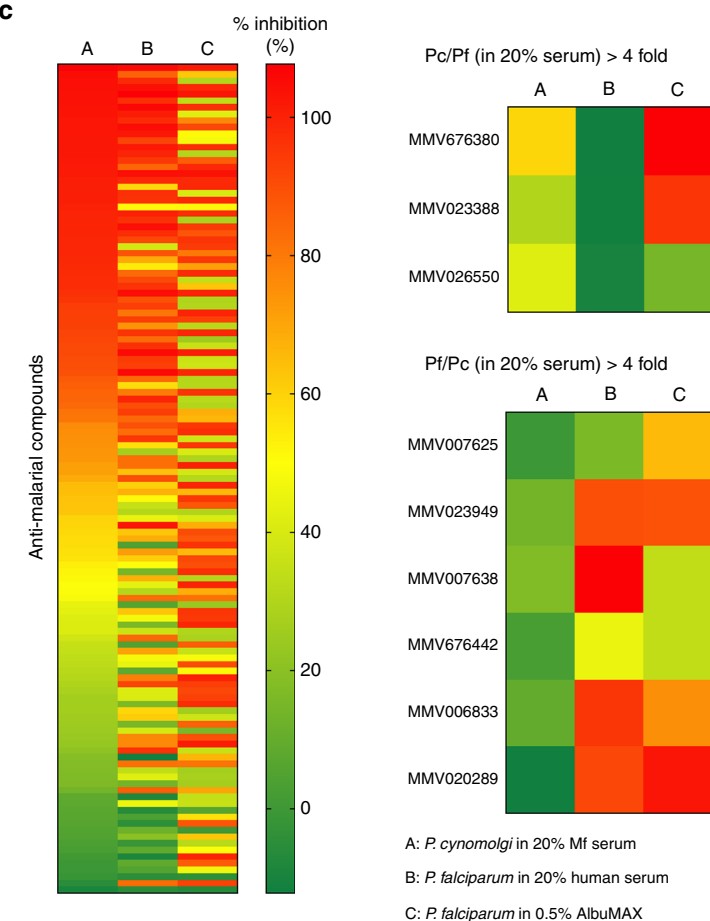

A: *P. cynomolgi* in 20% Mf serum

B: *P. falciparum* in 20% human serum

C: *P. falciparum* in 0.5% AlbuMAX

parasite species (*P. cynomolgi* and *P. falciparum*) used for the validation. Thus, it is likely that activity data from *P. cynomolgi*-based screens would be a more reliable predictor of activity against *P. vivax* than those that rely on *P. falciparum* or on the two most often used parasites of rodents *P. berghei* and *P. yoelii*[14]. Of late, various screening assays for *P. vivax* and *P. falciparum* liver stages has been developed to evaluate novel compounds[13,37–39]. Indeed, hypnozoites responsible for relapses, a major obstacle to successful control of *P. vivax*, are formed by *P. cynomolgi*, but not by any of the rodent parasite species. The urgent need to find an alternative to the anti-hypnozoitocidal 8-aminoquinoline compounds, whose potential toxicity impedes

**Fig. 4** Drug susceptibility testing using *P. cynomolgi* Berok K4 in vitro culture. **a** Correlation of *P. cynomolgi* Berok K4 and *P. falciparum* IC$_{50}$ values of common antimalarial reference compounds in a SYBR green I proliferation assay. The potency of the compounds was comparable between the two species, except for artemisinin, atovaquone and artesunate which were more potent in *P. cynomolgi* as compared with *P. falciparum*. A X = Y line indicates equal inhibition towards compound. **b** Heatmap showing IC$_{50}$ (μM) of a representative set of compounds from the Malaria Box. Majority of the compounds showed activity against both *P. falciparum* and *P. cynomolgi*, except for six compounds—MMV008127, MMV006250, MMV008270, MMV000563, MMV007839 and MMV008294 which displayed an IC$_{50}$ > 10 μM for *P. cynomolgi*, and < 5 μM for *P. falciparum*. **c** Heatmap showing percentage of inhibition of the 125 antimalarial compounds from the Pathogen Box in *P. cynomolgi* Berok K4 and *P. falciparum* in different serums and concentrations. Nine compounds (MMV676380, MMV023388, MMV026550, MMV007625, MMV023949, MMV007638, MMV676442, MMV006833 and MMV020289) showed more than fourfold difference in inhibition between *P. cynomolgi* Berok K4 and *P. falciparum* in their equivalent serums

widespread use, led to the development of an in vitro-based *P. cynomolgi* hepatic-stage assay[11,12]. One of the major logistical challenges that precludes high-throughput screening is the availability of infective sporozoites that can only be obtained from mosquitoes fed on blood collected from *P. cynomolgi*-infected macaques. The fact that the in vitro-maintained Berok K4 line parasites retained infectiousness to mosquitoes subsequent to cultivation augurs well for the eventual use of in vitro cultures to support mosquito infections. We are currently exploring diverse modifications to our culture protocols aimed at promoting gametocytogenesis, a process known to be highly sensitive to culture conditions, as was shown for *P. falciparum*[40]. This would minimise non-human primate use, and substantially increase opportunities for routine and regular sporozoite production.

Finally, it is important to point out that the *P. cynomolgi* model uniquely provides the possibility to confront fundamental biological, immunological and pathological insights, as well as therapeutic approaches derived from in vitro observations with the realities of experimental in vivo infections in the natural host. The ability to refine, or indeed to discard, a candidate compound(s) or vaccine formulation(s) through an in vitro–in vivo 'to-and-fro' significantly enhances the value of this pre-clinical model.

It is hoped that the availability of easily maintained in vitro erythrocytic *P. cynomolgi* parasites could now be exploited to conduct critical fundamental and translational research to develop drugs and vaccines against *P. vivax*, a widespread species whose control will determine the success of current efforts to eradicate malaria.

## Methods

**Ethical committees and animal welfare.** *Macaca fascicularis* (cynomolgus monkeys) were maintained at the Novartis Laboratory Animal Services, New Jersey, USA, (Novartis-LAS) and SingHealth Experimental Medicine Center, Singapore. Both institutions were audited and approved by the Novartis Animal Welfare Compliance. All animals were housed in accordance with the Guide for the Care and Use of Laboratory Animals and the Association for the Assessment and Accreditation of Laboratory Animal Care (AAALAC) Standards. All studies were approved by the Novartis Ethical Review Council and Novartis Institutional Animal Care and Use Committees prior (IACUC) to study initiation. In addition, work at SingHealth was approved by the SingHealth IACUC. The BPRC is an AAALAC-certified institute. Several rhesus macaques (*M. mulatta*) used in this study were captive bred for research purposes, and were housed at the BPRC facilities under compliance with the Dutch law on animal experiments, European directive 2010/63/EU and with the 'Standard for humane care and use of Laboratory Animals by Foreign institutions' identification number A5539–01, provided by the Department of Health and Human Services of the USA National Institutes of Health (NIH). Prior to the start of experiments at BPRC, all protocols were approved by the local independent ethical committee, according to Dutch law. Rhesus macaques were infected with $1 \times 10^6$ *P. cynomolgi* Berok K4 blood-stage parasites and bled at peak parasitaemia. About 300 female *Anopheles stephensi* mosquitoes Sind–Kasur strain Nijmegen (Nijmegen University Medical Centre St Radboud, Department of Medical Microbiology) were fed with this blood. Clinical samples utilised in this study were collected from *P. vivax*-infected malaria patients attending the clinics of the Shoklo Malaria Research Unit (SMRU), Mae Sot, Thailand, under the following ethical guidelines in the approved protocol: OXTREC 45–09 (University of Oxford, Centre for Clinical Vaccinology and Tropical Medicine, UK) and MUTM 2008–215 from the Ethics committee of Faculty of Tropical Medicine, Mahidol University.

**P. cynomolgi B, M and Berok strains.** Three *P. cynomolgi* strains were used in this study, the Berok strain[41], the B strain (*P. cynomolgi bastianelli*)[42] and M strain (*P. cynomolgi* Mulligan strain)[43,44]. The "B" strain or *P. cynomolgi bastianelli* was isolated by Garnham in 1959 from an infected *M. fascicularis* (previously named *M. irus*) near Kuantan, in the East Coast of Malaysia[42]. The *P. cynomolgi* Berok strain originated from Perak, Malaysia, where it was isolated from an infected *M. nemestrina* monkey[41]. The Berok strain sample that was used to derive the Berok K4 line consisted of a single cryopreserved 1 -mL sample passage through *Aotus* monkeys and dated as 2003. The sample was thawed out as described below, and washed prior to a tail–vein administration to a splenectomised *M. fascicularis* monkey. The *P. cynomolgi* 'M' strain was first described by Mulligan in 1935[44], and later shown to be transmitted to man 1961[43]. It has been suggested that the B and M strain isolates have been mixed-up at some time in the past, and that some of the B and M strain aliquots in use might actually have the same parasite line[3]. All strains of *P. cynomolgi* ultimately originated from stock collections at the CDC, which were then since propagated in monkeys at the recipient laboratories. In this study, the B and M strain were provided by B.R., the Berok strain cryovial used to generate the K4 line was provided by D.E.K.

**P. cynomolgi base medium composition for continuous culture.** The RPMI-1640 (Roswell Park Memorial Institute) supplemented with GlutaMAX (Gibco # 61870–036) containing 30 mM HEPES (Sigma-Aldrich), 0.2% (w/v) D-glucose and 200 μM hypoxanthine (Calbiochem). In all, 20% (v/v) *M. fascicularis* serum (heat inactivated) was added prior to use. The medium was filter-sterilised over 0.22 -μm filter.

**Macaque blood and serum extraction and preparation.** *M. fascicularis* erythrocytes and serum were obtained from Singapore Health Services Pte Ltd. Blood was collected by venous puncture into lithium heparin vacutainers (Becton-Dickinson). The blood collected was adjusted to 50% haematocrit with the RPMI-1640 supplemented with 30 mM HEPES before passing through pre-equilibrated non-woven filters (Antoshin) to deplete leucocytes. The erythrocytes were pelleted by centrifugation at 1800 rpm (650 rcf) for 5 min at room temperature (RT), washed thrice in the RPMI-1640 with 30 mM HEPES and stored at 4 °C in the same medium at 50% haematocrit until use. For collection of *M. fascicularis* serum, the blood was collected by venous puncture into SST serum separation vacutainers (Becton-Dickinson), inverted gently five times and allowed to clot in a vertical position. The tubes were then centrifuged at 3000 rpm (1810 rcf) for 10 min at RT. The *M. fascicularis* serum supernatant was collected and heat inactivated for 1 h at 56 °C and stored at −20 °C until use.

**In vitro parasite growth in M. fascicularis RBCs and serum.** *P. cynomolgi* parasite cultures were started from 1 mL of stabilate cryopreserved samples. The frozen samples were thawed-out using the sodium chloride method[45]. Briefly, sequentially decreasing concentrations of NaCl were added starting with 0.2 mL of 12% (w/v) NaCl added slowly drop-wise with gentle mixing, incubated at RT for 5 min without shaking, and further diluted in 10 mL of 1.6% (w/v) NaCl added drop-wise with gentle mixing. The samples were pelleted by centrifugation at 1800 rpm (650 rcf) for 5 min at RT, the supernatant discarded and samples were subjected to a final wash in 10 mL of 0.9% NaCl added drop-wise with gentle mixing, pelleted by centrifugation at 1800 rpm (650 rcf) for 5 min at RT and re-suspended in complete medium comprising *P. cynomolgi* base medium supplemented with 20% (v/v) macaque serum. Macaque blood was added to a 5% haematocrit. The parasites were cultured in plates in a modified candle jar (Stemcell technologies 27310) at 37 °C comprising about 3–5% $CO_2$, and 8–10% $O_2$[46,47]. Cultures were monitored daily by Giemsa-stained thick and thin smears by microscopy and parasitaemia adjusted between 0.5 and 5% for routine culture. The medium was changed daily. In order to eliminate the use of the candle jar, the cultures were adapted to trimix gas (5% $CO_2$, 5% $O_2$, 90% $N_2$). All other conditions were the same as for the modified candle jar. For culture in trimix gas, the parasites were gassed and cultured in non-vented flasks.

**Parasite synchronisation.** The infected RBCs from in vitro *P. cynomolgi* Berok K4 were first synchronised with 5% sorbitol, and then matured to schizonts which

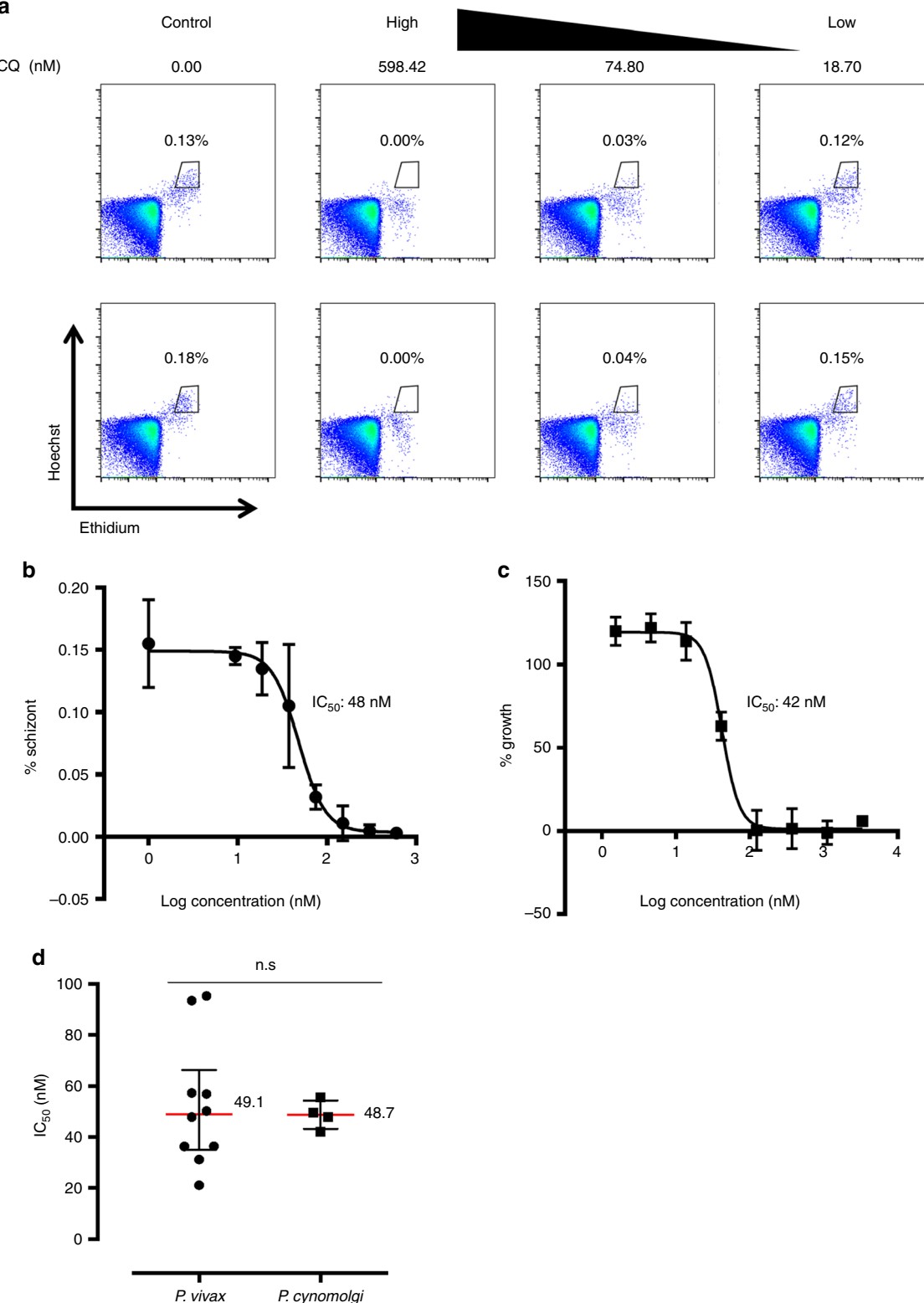

**Fig. 5** Correlation of chloroquine $IC_{50}$ of *P. cynomolgi* Berok K4 continuous culture with *P. vivax* clinical isolates. **a** Flow cytometry dot plots (Ethidium/Hoechst) of chloroquine-treated *P. cynomolgi* Berok K4 continuous culture gated for schizonts population. **b** $IC_{50}$ determination of chloroquine in *P. cynomolgi* Berok K4 continuous culture using the schizont maturation assay ($n = 3$) with error bars representing standard deviation (SD). **c** $IC_{50}$ determination of chloroquine in *P. cynomolgi* Berok K4 continuous culture using SYBR green I proliferation assay ($n = 3$) with error bars representing standard deviation (SD) **d** $IC_{50}$ of chloroquine in *P. cynomolgi* Berok K4 continuous culture and *P. vivax* clinical isolates using the schizont maturation assay. The median ($+/-$ IQR) values for the $IC_{50}$ of *P. vivax* clinical isolates and of *P. cynomolgi* Berok K4 continuous culture were similar. The non-parametric data in **d** was analysed using the Mann–Whitney U-Test with the significance level set at $P < 0.05$. The histograms and lines on box plots and scatter plots represent medians (*P. vivax* $n = 10$, *P. cynomolgi* $n = 4$) and the error bars the interquartile range (IQR)

were enriched using MACS LD columns (Miltenyi Biotech Asia Pacific Pte Ltd.) on magnetic sorter, where the flow through containing rings and early trophozoites were discarded. The retained late-stage parasites were eluted from the column, washed three times with the RPMI-1640 supplemented with 30 mM HEPES and then returned to culture.

**Determination of parasitaemia via light microscopy.** Thick and thin film smears were prepared with 5 μL of packed red blood cells for Giemsa staining, parasitaemia was determined from the thin film under a light microscope with ×1000 magnifications.

**Transmission study from Berok K4 continuous culture.** Monkey infections and mosquito feedings were performed as previously described[48]. Briefly, cryopreserved in vitro-adapted *P. cynomolgi* Berok K4 parasites were thawed as described above, and one million parasites were used to infect one *M. mulatta* monkey via intravenous injection, while the monkey was under ketamine sedation. Parasitaemia was monitored through Giemsa-stained thin smears. Female *An. stephensi* mosquitoes were fed with infected monkey blood using a glass feeder system on days 12 and 13 post infection, after which the monkey was cured from blood-stage parasites with chloroquine (three daily intramuscular injections 7.5 mg/kg). Oocysts were counted in at least ten mosquitoes at day 7 after the infected blood meal. The remaining mosquitoes were given a second uninfected blood meal to promote sporozoite invasion of the salivary glands.

**Sporozoites from Berok K4 continuous culture.** As previously described[48], sporozoites were isolated from salivary glands of infected *An. stephensi* mosquitoes 18 days post-infected blood meal.

**Infectivity of Berok K4 sporozoites from continuous culture.** *M. mulatta* hepatocytes from different donors were isolated as previously described[48], and then seeded onto collagen-coated 384-well plates at a density of 28,000 hepatocytes/well in William's B medium as previously described[48]. Two days later, 20,000 sporozoites were added to each well. Infections were carried out in six replicates per experimental condition. Daily medium refreshments were performed, and the infected cells were fixed in 4% PFA after 6 days of sporozoite infection. Parasites were immunostained with anti-PcHsp70 antibodies and goat-anti-rabbit Alexa 568 red secondary antibodies (Invitrogen A11011), while the nuclei were stained with DAPI before they were visualised using the Operetta high content screening automated microscope. Image analysis was done using the analysis algorithm as designed in previous study[48] with a minor adaptation for the change in fluorescent label of the secondary antibody.

**Micropipette aspiration and RBC sphericity measurement.** The micropipette aspiration technique was modified from Hochmuth, 2000[49]. Briefly, 1 μL of packed red blood cells containing ~1% parasitaemia and suspended in 1 mL of PBS (1% BSA). The samples were mounted onto the Olympus IX71 Inverted Microscope. A borosilicate glass micropipette (~1.5 -μm inner diameter) was used to extract the cell membrane under a negative pressure at a pressure drop rate of 0.5 pa/s. The corresponding cell membrane deformation was monitored using a ×100 oil immersion lens. The cell membrane deformation was recorded using the QColor5 High Resolution Color CCD Digital Fire Wire Camera (Olympus) and processed by QCapture Pro 6.0 (Olympus). The cell membrane shear modulus was calculated using the hemispherical cap model[49]. To quantify the binding force between uninfected red cells and a iRBC (rosetting cells), a double-pipette aspiration method was used[50]. A rosette was held by a micropipette (diameter = 2.0 ± 0.2 μm). A second micropipette was used to aspirate the uninfected red cell at increased levels of pressure. The force (F) to detach the red cell from the iRBC was calculated as $F = \pi r^2 \times P$, where $r$ is the inner diameter of the second micropipette, and $P$ is the pressure required for cell detachment. The aspiration pressure was measured by a pressure transducer (P61 model, Validyne Engineering), and recorded by USB-COM Data logger (Validyne Engineering). The process was recorded using a Dual CCD Digital Camera DP80 (Olympus®) at one frame/s. Recorded images were analysed with CellSens Dimension (Olympus®).

**Atomic force microscopy.** Infected RBCs (iRBCs) were harvested at the trophozoite stage and processed as follows: 200 μl of blood media mixture were supravitally stained with 1 μL of DAPI for 15 min in an incubator and prepared as smears (unfixed and air dried) for Atomic Force Microscopy (AFM). At least 20 iRBCs from each isolate and cell type was scanned using AFM. The total area analysed per parasite was 15 μm² ($n = 15$ for each respective group, of 1 μm² area/infected red cell). We were able to conduct serial measurements (with AFM, then and with Giemsa) by using a copper microdisk grid (H7 finder grid, SPI Supplies, PA) attached underneath the glass slide allowing us to locate and image the same cell. These thin smears were first AFM scanned by a Dimension 3100 model with a Nanoscope IIIa controller (Veeco, Santa Barbara, CA) using the tapping mode. The probes used for imaging were 125 -μm long by 30 -μm wide single-beam shaped cantilevers (Model PPP-NCHR-50, Nanosensors) with tip radius of curvature of

5–7 nm. Images were processed using the Nanoscope 5.30 software (Veeco, Santa Barbara, CA).

**Electron microscopy.** Sorted cells coated on poly-lysine (Sigma) glass coverslips were fixed in 2.5% glutaraldehyde, washed and treated with 1% osmium tetroxide (Ted Pella Inc.) before critical point drying (CPD 030, Bal-Tec). Glass coverslips were sputter coated with gold in a high vacuum (SCD005 sputter coater, Bal-Tec) and imaged with a field-emission scanning electron microscope (JSM-6701F, JEOL) at an acceleration voltage of 8 kV[51].

**Image stream analysis of resetting.** We determined the percentage of rosetting of *P. cynomolgi* iRBC by image stream analysis using a method adapted from Lee et al.[52]. Briefly, iRBCs stained with Hoechst and dihydroethidium were suspended in PBS to a haematocrit of 2% were assayed with the ImageStream 100 (Amnis, Seattle, WA) fitted with a ×60 objective. At least 200 untreated parasites for each single-stain condition were used to create a compensation matrix. During screening, 10,000 parasites were acquired and gated using the technique adopted from Malleret et al.[53] along with the added selection of cells that were not singlets (one or more cells adhering to an infected cell staining positive for Hoechst). Analysis was performed with the IDEAS software (version 4.0).

**DNA extraction, PCR amplifications and sequencing.** As described previously[54], the DNA was extracted, and the genes *rbp1b*, *rbp2a* and *rbp2b* were compared between the *P. cynomolgi* Berok, *P. cynomolgi* B strain and *P. cynomolgi* M strain.

**Compound libraries.** The Pathogen Box (http://www.pathogenbox.org/), modelled after the Malaria Box[55], is an open-source library comprising of 400 diverse open-source compounds targeting against neglected tropical diseases. The Malaria Box was an open-source library (until December 2015), which comprised 400 diverse compounds with antimalarial activity. In addition, the three drug candidates (the phosphatidylinositol-4-OH kinase (PI(4)K) inhibitor KDU691[56], the imidazolo-piperazines KAF179[57] and the spiroindolone KAF246[58]) were included as additional controls.

**Compound plate preparation.** Test compounds were prepared by serially diluting a 10 mM compound stock threefold with DMSO for eight concentration points in the master compound plate. In all, 100% DMSO was used as the negative control, and 10 mM mefloquine was used as the positive control. A 1000-fold dilution (either 100 nL or 50 nL) of compound from the master plate were spotted onto a 96-well or 384-well assay plate, respectively using Mosquito® nanoliter dispenser (Cambridge, UK). The plates were then sealed with a removable foil seal using PlateLoc Thermal Microplate Sealer (Agilent) until use.

**In vitro high-throughput IC50 drug susceptibility assay.** There are various high-throughput screens based on cultured *P. falciparum*[25,59–63] or on the hepatic stages of the rodent parasites *P. berghei* or *P. yoelii*[57,61] described. Here, we adapt the SYBR green I proliferation assay as described by Plouffe et al. Briefly, 50 μL of the *P. cynomolgi* culture were dispensed manually into the 384-well plate at both final parasitaemia of 0.5% (for dose response assay) or 1% (for single point assay) and the haematocrit adjusted to 2.5%. The 384-well assay plates were incubated at 37 °C for 72 h in 5% CO₂, 5% O₂ and 90% N₂. After a 72 h incubation, 10 μL of lysis buffer, consisting of 5 mM EDTA, 20 mM Tris-HCl pH 7.5, 1.6% Triton X-100 and 0.16% Saponin, were added to each well, and the plate was incubated in the dark for 24 h at RT. The plate was read for fluorescence using BioTek Synergy™ 4 hybrid microplate reader (Vermont, USA) using a bottom read mode at excitation wavelength of 485 nm and emission wavelength of 528 nm.

**Robustness of in vitro IC50 drug susceptibility assay.** The quality of the assay was determined with the Z' value[64], which takes into account the signal dynamic range and the data variation associated with the signal measurements and is calculated as follow: Z' = 1 – ((3 SD of positive control + 3 SD of negative control)/ (average of positive control–average of negative control)). An assay with Z' ≥ 0.5 < 1 is considered robust. Three repeats were carried out from which the Z' values showed the robustness of the assay. Read fluorescence (bottom read), λex = 485 nm. λem = 528 nm. Fluorescence values were normalised based on maximum fluorescence signal values for DMSO-treated wells and the minimum fluorescence signal values for wells containing the highest concentration of negative control compound, 10 μM mefloquine.

**Schizont maturation assay using flow-cytometry analysis.** The maturation assay for *P. vivax* was carried out as described previously[65,66], while parasite growth was assessed by flow cytometry[53] (Fig. 5a). In all, 200 μL of tightly synchronised ring-stage culture of *P. cynomolgi* was dispensed manually in the 96-well compound plate at a final parasitaemia of 0.5% and haematocrit adjusted to 2%. The assay plates were incubated at 37 °C for around 44 h in 5% CO₂, 5% O₂ and 90% N₂ until mid-late schizonts stage (> 5 merozoites) was observed in the drug-free control wells via Giemsa staining. Each well was well mixed, and 20 μL was

harvested into a small curved-bottom tube (Micronic) before 0.5 µL of 1 mg/mL dihydroethidium (Sigma) and 1 µL of 800 µM of Hoechst 33342 (Sigma) was added and made up to 100 µL with PBS. The tubes were incubated at RT for 20 min, and 100,000 events were acquired with an Accuri C6 (BD Biosciences, USA). The data were analysed using FlowJo software (Tree Star Inc.).

**Statistical analysis.** The parametric data presented in Fig. 2c, d were analysed using the Welch's *t* test with the significance level set at $P < 0.05$. The histograms represent means, and the error bars the standard deviation (SD). The non-parametric data (all data sets failed the D'Agostino & Pearson normality test) presented in Fig. 3b, d, f and h and Fig. 5d were analysed using the Mann–Whitney U test with the significance level set at $P < 0.05$. The histograms and lines on box plots and scatter plots represent medians and the error bars the interquartile range (IQR). All analyses were carried out using GraphPad Prism™ 7 for windows (GraphPad Software Inc, USA).

**Reporting Summary.** Further information on research design is available in the Nature Research Reporting Summary linked to this Article.

## Data availability
The source data underlying Figs. 1b, 1c, 2c, 2d, 3a–h, 4a–c, 5b–d, Supplementary Figs. 1 and 4 are provided as a Source Data file. Any additional data that support the findings of this study may be requested from the corresponding authors.

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

## Acknowledgements
We thank the Medicines for Malaria Venture (MMV, Switzerland) for providing the Pathogen Box compounds and accompanying information. We also thank John Ferretti and Nicole Ecklof for their support with the *Macaca fascicularis* work in Novartis Laboratory Animal Services, New Jersey, USA. The Novartis Institute Tropical Diseases (NITD), Singapore and the University of Otago, New Zealand co-financed the project. B.R. was jointly funded by the Royal Society New Zealand Marsden Fund (UOO1710) and the Health Research Council of New Zealand e-ASIA Joint Research Program (17/678). P.B. was partly supported by the National University of Singapore Start-up funds. L.R. was partly supported by a grant from the National Medical Research Council (#OFIRG17Nov123). IDMIT (G.S., ND-B, RLG) infrastructure is supported by the French government Programme d'Investissements d'Avenir (PIA), under grant ANR-11-INBS-0008 (INBS IDMIT). G.S. was supported by a grant from the Agence Nationale de la Recherche, France (ANR-17-CE13-0025-01). D.K. was supported by a grant from MMV (MMV RD/16/1082). SMRU is part of the Mahidol Oxford University Research Unit, supported by the Wellcome Trust of Great Britain.

## Author contributions
A.C.Y.C., D.E.K., B.R. and P.B. designed the project and experiments. A.C.Y.C., J.J.Y.O., R.S., V.K., B.M., A.M.Z., C.A.C, A.E., C.J.J, R.Z., B.H.T., S.N.A., A.Y., S.P.M, M.R.G, J.S.C., K.S.W.T., K.B., S.B., A.L., F.N., C.B., B.K.S.Y., D.M., N-D.B., R.L.G., G.S., B.R. and P.B. developed and performed the experiments, as well as analysed the data. A.C.Y.C., D.E.K., B.K.S.Y., C.H.M.K., L.R., T.T.D., G.S., B.R. and P.B. developed and designed the concept and provided resources. A.C.Y.C., B.R., G.S. and P.B. wrote the paper, and all authors reviewed and edited the paper.

## Additional information

**Competing interests:** A.C.Y.C., J.J.Y.O., B.H.T., S.N.A., A.Y., K.B., S.B., A.L., C.B., B.K.S.Y., T.T.D. and P.B were/are employed by and/or shareholders of Novartis Pharma AG. The remaining authors declare no competing interests.

