## [Peer Review File · Nature Communications]

Editorial Note: This manuscript has been previously reviewed at another journal that is not operating a transparent peer review scheme. This document only contains reviewer comments and rebuttal letters for versions considered at Nature Communications .

Reviewers' Comments:

Reviewer #1:

Remarks to the Author:

The authors have responded to comments and concerns. Unfortunately, they do not provide any substantive new data in response to this and other reviewers' concerns. Importantly, they do not show that the asexual blood stage culture system of cynomolgi supports the generation of gametocyte stages and allows mosquito infections from cultured parasites. Gametocyte culture of course would be the basis to routinely produce sporozoites and enable drug screening against dormant liver stages, the important platform need for malaria drug development.

The authors argue that the asexual blood stage culture system in itself is a transformational advance as it enables the screening of drugs for the related human pathogen vivax and frame this in their response as a critical need to the field. This reviewer disagrees. In reality, drug screening, discovery and development for malaria parasite asexual stages is and will remain to be focused on falciparum, the most lethal malaria parasite. Any drug that will be successful against falciparum will then be used to treat vivax infections. That is what happened in the past and that is what will happen in the future. Certainly, there might be differences in drug susceptibility and the acquisition of resistance between vivax and falciparum but this reviewer doubts very much that this will result in a stand-alone asexual drug discovery effort for vivax.

REVIEWERS' COMMENTS:

Reviewer #1 (Remarks to the Author):

The authors have responded to comments and concerns. Unfortunately, they do not provide any substantive new data in response to this and other reviewers' concerns. Importantly, they do not show that the asexual blood stage culture system of *cynomolgi* supports the generation of gametocyte stages and allows mosquito infections from cultured parasites. Gametocyte culture of course would be the basis to routinely produce sporozoites and enable drug screening against dormant liver stages, the important platform need for malaria drug development.

The authors argue that the asexual blood stage culture system in itself is a transformational advance as it enables the screening of drugs for the related human pathogen *vivax* and frame this in their response as a critical need to the field. This reviewer disagrees. In reality, drug screening, discovery and development for malaria parasite asexual stages is and will remain to be focused on *falciparum*, the most lethal malaria parasite. Any drug that will be successful against *falciparum* will then be used to treat *vivax* infections. That is what happened in the past and that is what will happen in the future. Certainly, there might be differences in drug susceptibility and the acquisition of resistance between *vivax* and *falciparum* but this reviewer doubts very much that this will result in a stand-alone asexual drug discovery effort for *vivax*.

Response to Reviewer 1 comments:

We fully agree that it would be ideal if the *P. cynomolgi* Berok strain adapted to continuous culture described here would produce a higher number of sporozoites and hypnozoites *in vitro*. Unfortunately, our preliminary result shows that indeed this strain produces lesser numbers of hypnozoites than the B or M strains. Presently, the genes involved in hypnozoites production remain unknown and this is why we strongly believe that putting this article and strain out there, in the malaria community, would allow for other labs to participate in expanding our understanding of the Plasmodium liver stage cycle. This would accelerate investigation into deciphering the biology of hypnozoites formation and maturation. Our laboratories are presently attempting to genetic cross the strain described here with the B strain which is known to produce higher numbers of hypnozoites. This is an effort that, if successful, will take at least one or two years to achieve, and hence an unjustifiable amount of time to hold this work back.

Our work on asexual blood stage as described here fulfills two purposes: the first is to illustrate how robust our culture is exemplified by its adaptability to high-throughput drug screening and the second, to highlight *P. cynomolgi* and *P. vivax* share many common characteristics including susceptibility patterns which can be different for those of *P. falciparum*. Our laboratory, in collaboration with Medicines for Malaria Venture, is currently investigating the mechanism of resistance of new drug candidates which have proven to have strong activity against *P. falciparum* but not *P. vivax*, for instance DSM265 (see Alejandro Llanos-Cuentas *et. al.*, Lancet Infect Dis 2018; 18: 874–83).